

# The ability of Reciproc instruments to reach full working length without glide path preparation: a clinical retrospective study

Andreas Bartols[1,2], Bernt-Peter Robra[3] and Winfried Walther[1]

[1] Dental Academy for Continuing Professional Development Karlsruhe, Karlsruhe, Germany
[2] Clinic for Conservative Dentistry and Periodontology, Christian-Albrechts-University Kiel, Kiel, Germany
[3] Institute of Social Medicine and Health Economics, Otto-von-Guericke University Magdeburg, Magdeburg, Germany

## ABSTRACT

**Background**. Reciproc instruments are the only contemporary root canal instruments where glide path preparation is no longer strictly demanded by the manufacturer. As the complete preparation of root canals is associated with success in endodontic treatment we wanted to assess the ability and find predictors for Reciproc instruments to reach full working length (RFWL) in root canals of maxillary molars in primary root canal treatment (1°RCTx) and retreatment (2°RCTx) cases.

**Methods**. This retrospective study evaluated 255 endodontic treatment cases of maxillary molars. 180 were 1°RCTx and 75 2°RCTx. All root canals were prepared with Reciproc instruments. The groups were compared and in a binary logistic regression model predictors for RFWL were evaluated.

**Results**. A total of 926 root canals were treated with Reciproc without glide path preparation. This was possible in 885 canals (95.6%). In 1°RCTx cases 625 of 649 (96.3%) canals were RFWL and in 2°RCTx cases 260 of 277 (93.9%). In second and third mesiobuccal canals (MB2/3) 90 out of 101 (89.1%) were RFWL with Reciproc in 1°RCTx and in the 2°RCTx treatment group 49 out of 51 cases (96.1%). In mesio-buccal (MB1) canals "2°RCTx" was identified as negative predictor for RFWL (OR 0.24 (CI [0.08–0.77])). In MB2/3 canals full working length was reached less often (OR 0.04 (CI [0.01–0.31])) if the tooth was constricted and more often if MB2/3 and MB1 canals were convergent (OR 4.60 (CI [1.07–19.61])).

**Discussion**. Using Reciproc instruments, the vast majority of root canals in primary treatment and retreatment cases can be prepared without glide path preparation.

## INTRODUCTION

Preparation of root canals in a way that they can be completely disinfected is a big challenge in endodontic treatment and retreatment. Technical difficulties are common during negotiating and preparing root canals as naturally the anatomy of root canals has a great variability (*Briseno-Marroquin et al., 2015*; *Vertucci, 1984*) and, especially in maxillary

Corresponding author
Andreas Bartols,
andreas_bartols@azfk.de

molars, often second mesiobuccal (MB2) and, sometimes even third mesiobuccal (MB3) canals are present (*Kulild & Peters, 1990*; *Schwarze et al., 2002*). These MB2/3 canals are smaller than the main canals, often covered with dentine (*Zuolo, Carvalho & De-Deus, 2015*), obliterated and curved. Moreover, the other tooth structure is often calcified (*Amir, Gutmann & Witherspoon, 2001*; *McCabe & Dummer, 2012*). In retreatment cases more obstacles to root canal negotiation will arise, as old root canal filling materials have to be removed and procedural errors from initial treatment must be corrected (*Gluskin et al., 2008*) or cannot be corrected.

The latest developments in mechanical root canal preparation led to the introduction of reciprocating single file techniques (*Bürklein et al., 2012*; *Yared, 2008*). Rotary nickel-titanium (NiTi) multiple file systems have already enhanced the technical quality of root canal preparation (*Schäfer, Schulz-Bongert & Tulus, 2004*). But for all rotary instruments, coronal enlargement of the canal and the preparation of a glide path was demanded (*Berutti et al., 2009*; *West, 2010*). The glide path is defined as a smooth tunnel from the root canal orifice to the apical ending of the root canal (*West, 2010*). A glide path can be prepared with stainless steel hand-instruments of small sizes like ISO 06, 08 and 10, going up to ISO 20. The pre-established tunnel reduces the stress on the tips of rotary root canal instruments and therefore also reduces the risk of instrument fractures. With the introduction of single-file and single-use Reciproc instruments (VDW, Munich, Germany) the manufacturer claimed that the glide path preparation with these instruments generally is no longer mandatory (*Yared, 2011*) even not in calcified MB2 canals (*Yared, 2013a*). This was explained by the special reciprocating movement of the instruments meaning that the instruments cut dentine in a counter-clockwise (CCW) direction and are immediately released in a clockwise (CW) motion. As the CW rotation is smaller than the CCW rotation, the instrument will advance into the root canal. The releasing motion takes the stress from the instruments and prevents, in combination with a new reinforced M-wire NiTi-alloy, instrument fractures. Because of the reciprocating motion, the instruments should follow the naturally existent root canal path down to the apical ending of the canal (*Yared, 2011*). That indeed a large proportion of root canals can be prepared with Reciproc without previous glide path preparation was recently confirmed (*De-Deus et al., 2013*; *Zuolo, Carvalho & De-Deus, 2015*).

In retreatment cases different techniques for gutta-percha removal were advocated. This involves the application of hand files, NiTi rotary files, ultrasonic (US) instruments, heat or solvents (*Ferreira, Rhodes & Ford, 2001*; *Wilcox et al., 1987*). Also special rotary NiTi retreatment files were developed to enable gutta-percha removal (*Rödig et al., 2012*). However, these instruments have the sole purpose of efficiently removing the bulk of the old obturation materials. After that, glide path preparation remains mandatory to prepare the whole root canal (*Gluskin et al., 2008*). The manufacturer of Reciproc instruments also claimed that retreatment cases can be safely solved with these instruments as long as the manufacturer's instructions are followed (*Yared, 2013b*). In several *in-vitro* studies this claim was confirmed (*De Souza et al., 2015*; *Marfisi et al., 2015*; *Zuolo et al., 2013*). However, only completely filled root canals were retreated in these studies. Therefore the question remains if the procedure remains as safe as believed in clinical reality with

incompletely filled canals with unexpected blockages, ledges and other obstacles, and if in these situations root canal preparation can be done without glide path preparation.

Therefore we analysed all our endodontic treatment cases of maxillary molars with Reciproc instruments retrospectively. We sought to answer the question if in primary and also in retreatment cases the technical root canal preparation can be carried out with Reciproc instruments without glide path preparation to the full working length. Moreover, we wanted to reveal predictors for reaching or not reaching full working length. Our hypothesis was that the type of treatment (primary vs. retreatment cases), constricted root canals and convergent MB2/3 and MB1 canals are predictors for reaching full working length with Reciproc instruments.

## MATERIALS AND METHODS

For this retrospective study, patient files from the outpatient clinic of the Dental Academy of Continuing Professional Development Karlsruhe, Germany were used. Data were collected without reference to patient names and completely anonymized for evaluation. Because of the retrospective data collection, this study was a non-intervention clinical trial and did not interfere with the psychological or physical integrity of patients. The study was conducted according to the European guidelines for good clinical practice (CPMP/ICH/135/95) and according to the Professional Code for Physicians of the Medical Council of the State of Baden-Württemberg. The Institutional Review Board of the Baden-Württemberg Medical Council reviewed the study from the ethical perspective and approved it (AZ: F-2016-031-z).

### Study sample

For this clinical retrospective study all maxillary molars that were treated endodontically with Reciproc from October 2011 to October 2015 were identified. All treatments were performed by one single operator (AB) with ten years of extensive operative experience in endodontics. All primary root canal treatment cases (1°RCTx) and all orthograde root canal retreatment cases (2°RCTx) were included. The age of the patients was 16 years or older. Teeth with incomplete root development and retreatment cases with previously performed apicoectomies were excluded. However, no further exclusion criteria were set regarding canal curvature, radiographically narrow canals or preoperative restoration of the tooth.

The following information was collected from the medical records for every individual case: gender of the patient, tooth number, quantity of root canals, type of treatment (1°RCTx/2°RCTx), constricted root canal (yes/no) separately for every canal (first mesiobuccal (MB1), second/third mesiobuccal (MB2/3), distobuccal (DB) and palatal (P)), reaching full working length (RFWL) or not reaching full working length (NRFWL) separately for every canal, a glide path was prepared (yes/no) separately for every canal, MB2/3 and MB1 canals convergent (yes/no), apical perforation (yes/no), Forfenan retreatment case (yes/no), Reciproc instrument fracture (yes/no) and apicoectomy after orthograde (re-)treatment (yes/no).

## Treatment procedures

Generally all patients were treated under local anaesthesia. All treatments were performed according to the quality guidelines of the *European Society of Endodontology (2006)*. In every case a rubber dam was used and the complete treatment was done with the use of a dental operating microscope (DOM) (PROergo S7; Zeiss, Jena, Germany). For all preparations a VDW.SILVER Reciproc motor (VDW) in combination with a RootZX apex locator (J. Morita Europe GmbH, Dietzenbach, Germany) or a VDW.GOLD Reciproc motor with integrated apex locator (VDW) was used with the preset program "RECIPROC ALL". Before treatment the preoperative radiographs were used to measure the approximate root canal lengths. The precise working lengths were determined during root canal preparation with the apex locator; for this the complete root canal length was measured and 0.5 mm subtracted to set the WL.

## Primary root canal treatment procedures

Straight line access to the root canal orifices was established with diamond coated burs and Miller burs (both Komet Brasseler, Lemgo, Germany). All canal orifices including the MB2/3 orifices were opened with an EndoGuide bur EG7 (SS White Burs, Inc., Lakewood, NJ, USA). In case of deeply calcified root canals the orifices were further prepared with a diamond coated ultrasonic tip (3E Tip on Tigon; W&H, Bürmoos, Austria). After that a Reciproc R25 was used to preflare the coronal two-thirds of the root canal. The Reciproc instruments were used strictly according to the manufacturers recommendations for root canal preparation without glide path (*Yared, 2011*) and according to the special recommendations for MB2 canals (*Yared, 2013a*). During preparation, the canals and pulp chamber were flooded with 3% sodium hypochlorite (NaOCl). After enlargement of the coronal half of the root canal, the canal was scouted with a size 06 C-Pilot file (VDW). If the instrument could not be easily advanced further into the canal without resistance, the canal was rated as "constricted". When the R25 reached two-thirds of the canal length, the working length (WL) was determined using a size 10 C-Pilot file (VDW) with an apex locator. When the C-Pilot file could not reach WL, the R25 was used again to work in the canal. The procedure was repeated until WL could be determined passively with the C-Pilot file. Afterwards the complete WL was prepared with the R25 and the preparation was classified as RFWL. When the R25 did not advance further into the root canal, it was tried to actively prepare a manual glide path with if necessary pre-bent 06, 08, 10 and 15 C-Pilot files as a matter of principle. The preparation was then classified as NRFWL with the R25. The aim was in each case to gain apical patency with a 10 C-Pilot file.

In cases of large root canals, e.g., palatal (P) canals, an additional R40 or R50 was used for the preparation of that canal. All instruments were used in one molar and were afterwards discarded. If signs of deformation were visible at the instruments, they were immediately discarded and replaced by a new instrument. All canals were further instrumented with NiTi hand files (VDW) to at least ISO 35 to prevent locking of the 30 gauge Flexi-Glide Utility Tip (Vista Dental Products, Racine, WI, USA) that was used for irrigation and to allow the deepest possible placement of the irrigation tip towards the end of the root canal preparation. After completed instrumentation, all canals were irrigated with EDTA 15%

and afterwards again disinfected with NaOCl 3%. All solutions were used with passive ultrasonic irrigation (PUI) (Irrisafe on P5 Newtron, Acteon Germany GmbH, Mettmann, Germany). After that, calcium hydroxide (Ca(OH)$_2$) was placed as root canal dressing or in case of single visit treatment a gutta-percha root canal filling was placed. The detailed treatment protocol was described in an earlier publication (*Bartols, 2013*).

## Orthograde root canal retreatment procedures

In retreatment cases the same instruments were used for access cavity preparation as in 1°RCTx cases. In case of a MB2/3 canal, the MB1 canal was prepared first using a R25 file. In cases of hard obturation materials, they were initially centre punched with a #1 Gates bur (Komet Brasseler, Lemgo, Germany) for easier advancement of the R25. The R25 was used according to the 1°RCTx procedures. At first all filling material was removed. After that, according to the 1°RCTx cases, the remaining of the canal was scouted with a size 06 C-Pilot file and was rated as "constricted" if the instrument could not easily advance into the apical part of the canal. All MB2/3 canals were not prepared previously and therefore constriction was determined according to 1°RCTx cases. When about two-thirds of the WL was prepared, the root canal walls were cleaned in a brushing motion with the R25 to remove as much as possible of the old obturation material. Then, an attempt was made to introduce a 10 C-Pilot file to determine the working length with the apex locator. If this was not possible, the R25 was used again, until passive negotiation with the 10 C-Pilot file was possible. When the WL was determined, the complete length was prepared with the R25 and the preparation classified as RFWL. When a glide path had to be created, the preparation was classified analogue to the 1°RCTx group as NRFWL. Apical patency was maintained with a 10 C-Pilot file. The cleanliness of the root canal walls was inspected under high magnification in the DOM. Remnants of obturation materials were removed with an Endo file K15 or K25 on the P5 Newtron (both Acteon).

The further treatment and disinfection protocol was the same as with 1°RCTx group. Also for retreatment cases a detailed protocol was described in an earlier publication (*Bartols, 2013*).

## Statistical analyses

SPSS (Version 21, Win x64, IBM, Armonk, New York, USA) was used for all statistical analyses. The distributions to the different treatment groups were compared with the Pearson-chi-square test. The α-type error was set to 0.05.

Binary logistic regression analyses were performed to take potential factors that affect RFWL into account simultaneously. From clinical experience and theoretical considerations, we hypothesized that potential factors that would affect RFWL could be the type of treatment (1°RCTx vs. 2°RCTx), constricted canals (yes/no) and in case of MB2/3, if the canal was convergent with MB1.

## RESULTS

We identified 255 maxillary molars that met the inclusion criteria. 180 were primary and 75 secondary treatment cases. Tooth and root canal distributions are summarized in Table 1.

**Table 1  Frequencies of tooth types, root canal distributions and type of treatment.**

|  | Quantity of teeth | Quantity of root canals per tooth | | | | | | Type of treatment | | | |
|---|---|---|---|---|---|---|---|---|---|---|---|
|  |  | **3** | | **4** | | **5** | | **1°RCTx** | | **2°RCTx** | |
| First maxillary molar *N* (%) | 153 | 37 | (24.2) | 107 | (69.9) | 9 | (5.9) | 101 | (66.0) | 52 | (34.0) |
| Second maxillary molar *N* (%) | 102 | 66 | (64.7) | 36 | (35.3) | 0 | (0.0) | 79 | (77.5) | 23 | (22.5) |
| Total *N* | 255 | 103 | (40.4) | 143 | (56.1) | 9 | (3.5) | 180 | (70.6) | 75 | (29.4) |

**Notes.**
1°RCTx, primary root canal treatment; 2°RCTx, orthograde endodontic retreatment.

We found 152 (59.6%) MB2/3 canals in 255 maxillary molars. 107 (69.9%) second mesiobuccal and nine (5.9%) third mesiobuccal canals in 153 first maxillary molars and 36 (35.3%) MB2 canals in 102 second maxillary molars. Overall 926 root canals were approached to be prepared without prior glide path preparation (Table 2). This was possible in 885 cases, an overall rate of 95.6%. In 1°RCTx cases 625 of 649 (96.3%) canals were RFWL and in 2°RCTx cases 260 of 277 (93.8%).

There were differences in the two treatment groups 1°RCTx and 2°RCTx in the ability of Reciproc instruments to reach full working length (Table 2). A total of 175 out of 180 (97.2%) MB1 canals were prepared with Reciproc to full WL in 1°RCTx cases. In the 2°RCTx treatment group this was possible in 67 out of 75 cases (89.3%). The difference was statistically significant ($X^2 = 6.81$; $p = 0.009$). In the five NRFWL 1°RCTx canals an attempt was made to prepare a manual glide path because the R25 did not advance further into the root canal. This was not possible and therefore two canals were prepared incompletely, because these canals were not patent at any time during the preparation procedures. In the three other canals the active glide path preparation was also not possible. In these canals, the further use of the R25 without glide path preparation resulted in an apical perforation. In one of these roots the apical perforation was corrected with an apicoectomy. In one case, the tooth was extracted because the patient decided against perforation repair and a surgical intervention and in the other case the patient decided to do nothing, because he was clinically symptom-free. In five out of eight canals in the 2°RCTx group, the R25 did not reach full WL and also the manual glide path preparation was not possible. These canals were prepared incompletely and were never patent during preparation procedures. In the three other canals, the preparation without glide path ended in an apical perforation. These three apically perforated roots were treated surgically.

A total of 180 out of 180 (100.0%) DB canals were fully prepared with Reciproc instruments in 1°RCTx cases. In the 2°RCTx treatment group this was possible in 69 out of 75 cases (92.0%). The difference was statistically significant ($X^2 = 14.75$; $p < 0.001$). In the 2°RCTx group a manual glide path preparation was not possible in four out of six cases in the DB canal. One tooth had to be treated surgically. In the three remaining canals the use of R25 without glide path resulted in an apical perforation. Two of these cases were treated surgically.

All 255 palatal root canals were prepared to full WL regardless of the treatment group.

A total of 90 out of 101 (89.1%) MB2/3 canals were completely prepared with Reciproc in 1°RCTx cases. In the 2°RCTx treatment group this was possible in 49 out of 51 cases

Bartols et al. (2017), *PeerJ*, DOI 10.7717/peerj.3583

**Table 2  Root canals evaluated as reaching full working length (RFWL).**

| Type of treatment | Tooth type | Total teeth N | Total root canals N | MB1 RFWL N (%) | Total MB1 RFWL % | DB RFWL N (%) | Total DB RFWL % | P RFWL N (%) | Total P RFWL % | Total MB2/3 canals N | MB2/3 RFWL N (%) | Total MB2/3 RFWL % | Total canals RFWL (%) |
|---|---|---|---|---|---|---|---|---|---|---|---|---|---|
| 1°RCTx | First maxillary molar | 101 | 389 | 100 (99.0) | | 101 (100.0) | | 101 (100.0) | | 78 | 67 (85.9) | | 369 (94.9) |
| | Second maxillary molar | 79 | 260 | 75 (94.9) | 97.2[a] | 79 (100.0) | 100.0[a] | 79 (100.0) | 100.0[a] | 23 | 23 (100.0) | 89.1[a] | 256 (98.5) |
| 2°RCTx | First maxillary molar | 52 | 195 | 47 (90.4) | | 49 (94.2) | | 52 (100.0) | | 38 | 37 (97.4) | | 185 (94.9) |
| | Second maxillary molar | 23 | 82 | 20 (87.0) | 89.3[b] | 20 (87.0) | 92.0[b] | 23 (100.0) | 100.0[a] | 13 | 12 (92.3) | 96.1[a] | 75 (91.5) |
| | Total **N** | 255 | 926 | 242 (94.9) | | 249 (97.6) | | 255 (100.0) | | 152 | 139 (91.4) | | 885 (95.6) |

**Notes.**

1°RCTx, primary root canal treatment; 2°RCTx, orthograde endodontic retreatment; MB, mesiobuccal root canal; DB, distobuccal root canal; P, palatal root canal.

Values with different superscript letters indicate statistically significant differences in columns (Pearson-chi-square test, $p < 0.05$).

(96.1%). The difference was not statistically significant ($X^2 = 2.105$; $p = 0.147$). In eight cases of the eleven 1°RCTx cases NRFWL, a manual glide path preparation was attempted. By that, seven cases were solved. In one case this was not possible and the canal was ultimately classified NRFWL. In the three remaining canals an apical perforation occurred with R25. In one of these teeth this was corrected with an apicoectomy. In the other cases the patients decided to do nothing, because they were clinically symptom-free. In one case a R25 file separated during MB2 preparation. The instrument was removed and the canal was prepared with a new R25 without glide path preparation. In the 2°RCTx group in one case the MB2 was prepared completely after manual glide path preparation. In the other case a R25 fractured. The fragment could not be removed. The patient was clinically symptom-free and decided to leave the instrument *in situ*.

In the 152 teeth with an MB2/3 canal, 51 canals were classified as constricted. In twelve (23.5%) of the 51 constricted cases R25 was NRFWL, while in only one (1.0%) of the other 101 not constricted cases R25 was NRFWL. The difference was statistically significant ($X^2 = 22.012$; $p < 0.001$).

Of the 152 MB2/3 canals, 90 were classified as convergent with MB1 and 60 as not convergent. For the remaining two cases the data was missing in the medical files. In three (3.3%) of the convergent cases full WL was not reached with R25. In ten (16.7%) of the not convergent cases full WL was not reached with R25. The difference was statistically significant ($X^2 = 8.085$; $p = 0.004$).

The 2°RCTx group included 9 Forfenan ("Russian-Red-Cement") retreatment cases. In seven of these cases R25 reached full WL. In one of the two other cases a manual glide path preparation was performed ending in an apical perforation. In the other case an apical perforation resulted from repeated R25 preparation without glide path. The first mentioned tooth was extracted and the second tooth was treated with an apicoectomy.

During the preparation of 926 root canals in 255 teeth, two instruments fractured (in 0.2% of the canals and in 0.7% of the teeth) and 12 apical perforations (in 1.3% of the canals and in 4.7% of the teeth) occurred.

The binary logistic regression models for RFWL (yes/no) are summarized in Table 3. For MB2 canals we found a significant influence to RFWL of the covariates "constricted root canal" and "MB2/3 and MB1 convergent" (both $p < 0.05$). The chance was smaller in constricted root canals to RFWL and higher in teeth with convergent MB2/3 and MB1 canals.

For MB1 canals there was a significant association with the covariate "type of treatment" ($p < 0.05$). In retreatment cases the chance of reaching FWL was lower than in primary treatments. The covariate "constriction" was not significant in MB1 canals.

For DB canals we did not identify any covariates of significant influence. A regression model for the P canals could not be computed because all canals reached full working length.

## DISCUSSION

Our retrospective clinical study shows that a large proportion of demanding root canals was prepared without glide path preparation in maxillary molars. This is the case for primary

**Table 3** Binary logistic regression modeling for tooth related factors affecting RFWL.

| Root canal | Covariate | RFWL (%) | Odds ratio (95% CI) | p-Value |
|---|---|---|---|---|
| MB2/3 | **Type of treatment** | | | |
| $N = 150$ | 1°RCTx | 90 (89.1%) | 1 | |
| | 2°RCTx | 49 (96.1%) | 3.27 (0.60–17.85) | 0.172 |
| | **Constricted root canal** | | | |
| | Yes | 39 (76.5%) | 0.04 (0.01–0.31) | |
| | No | 100 (99.0%) | 1 | **0.002** |
| | **MB2/3 and MB1 convergent** | | | |
| | Yes | 87 (96.7%) | 4.60 (1.07–19.61) | |
| | No | 50 (83.3%) | 1 | **0.040** |
| | Nagelkerke $R^2 = 0.392$ | | | |
| MB1 | **Type of treatment** | | | |
| $N = 255$ | 1°RCTx | 175 (97.2%) | 1 | |
| | 2°RCTx | 67 (89.3%) | 0.24 (0.08–0.77) | **0.016** |
| | **Constricted root canal** | | | |
| | Yes | 68 (91.9) | 0.46 (0.15–1.45) | |
| | No | 174 (96.1%) | 1 | 0.187 |
| | Nagelkerke $R^2 = 0.091$ | | | |

**Notes.**
1°RCTx, primary root canal treatment; 2°RCTx, orthograde endodontic retreatment; RFWL, reaching full working length. Bold p-values indicate statistical significance of $p < 0.05$ in the logistic regression model.

root canal treatments and for retreatments. In our logistic regression model we identified "convergent MB2/3 and MB1 canals" as positive predictor for RFWL in MB2 canals and "root canal is constricted" as negative predictor. For MB1 canals we identified "2°RCTx" as negative predictor for RFWL.

Lately, a study identified "achieving patency at the root canal terminus" as an important prognostic factor for improved healing of periapical lesions (*Ng, Mann & Gulabivala, 2011b*) and for tooth survival (*Ng, Mann & Gulabivala, 2011a*). Therefore, it is of utmost importance that root canals are prepared to full WL so that proper disinfection is possible afterwards. In our study RFWL automatically included achieving patency with an ISO 10 C-Pilot file at the canal terminus. With rotary NiTi instruments glide path preparation is necessary to prepare root canals to the apical canal terminus to avoid instrument fractures and keep apical patency. Many instruments are available for glide path preparation and the procedure often needs a lot of patience, especially in cases with obliterated root canals (*West, 2010*) and can be substantially time consuming. Moreover, when manual glide path preparations are performed, dentists described higher general physical strain and strain on their fingers than with Reciproc instruments (*Bartols et al., 2016*). If the glide path preparation can be avoided this does not represent a primary biological objective, but may reduce the effort for root canal preparation in the dimension of treatment time, number of instruments and may reduce the physical strain for the operator.

It is difficult to find an appropriate comparison group for the question we sought to answer. At the moment there is only one machine driven instrument system (Reciproc) with which glide path preparation is no longer strictly recommended by the manufacturer. All

NiTi rotary instrument systems normally need a glide path preparation to avoid instrument fractures (*West, 2010*) that occur because of torsional stresses on the instrument (*Berutti et al., 2009*; *De Oliveira Alves et al., 2012*). From an ethical point of view, it is therefore not conceivable to compare rotary instruments without glide path preparation with Reciproc in a clinical experiment head-to-head. Of course, such experiments are possible in *in-vitro* studies. However, these studies do not completely cover the clinical reality with a lot more practical problems, especially in retreatment cases, in contrast to completely filled laboratory retreatment cases without obstacles like ledges, blockages and other problems. Therefore, we decided to choose a retrospective study design to evaluate a series of treatments that had been done anyway.

Overall we found a rate of 95.6% of root canals that were prepared with Reciproc instruments to full WL. This is a large proportion of canals that were treated without glide path preparation. An *in-vitro* study assessed the possibility to reach full WL with Reciproc instruments in straight and moderately curved root canals of mandibular molars which was possible in 96.4% and 90.7% respectively (*De-Deus et al., 2013*). Therefore, our overall rate of RFWL lies between these values. Interestingly in the study of *De-Deus et al. (2013)* in 98 root canals after coronal and middle third preparation of the root canals a size 10 file could not reach full WL while the repeated use of the R25 led to RFWL in 67.3% of these canals. Therefore, a lot of root canals in the aforementioned study were prepared completely with the R25 that otherwise would have been prepared incompletely.

There is only little information in literature on typical reference values for reaching full WL during root canal treatment in a clinical situation. Two connected papers (*Ng, Mann & Gulabivala, 2011a*; *Ng, Mann & Gulabivala, 2011b*) contain indirectly a proportion of root canals that were assessed as patent during root canal treatment. Therefore, we assume that these canals were prepared to full WL. For 1°RCTx the calculated rates from these papers are 93.5% and 94.6%, respectively, and for 2°RCTx 86.0% and 91.1% (*Ng, Mann & Gulabivala, 2011a*; *Ng, Mann & Gulabivala, 2011b*). However, there is no information about the types of root canals treated. Our overall rates of RFWL with Reciproc instruments are about 2–3% higher as these values (1°RCTx 96.3% and 2°RCTx 93.8%) and moreover contain very demanding situations in terms of the preparation of MB2/3 canals of maxillary molars. Moreover, only treatments of maxillary molars are evaluated in our study, while in the studies cited, every tooth type was included. An *in-vitro* study evaluated the R25 for RFWL in straight and moderately curved root canals in mandibular molars (*De-Deus et al., 2013*). For straight canals, a rate of 96.4% for RFWL was found and a rate of 90.7% for moderately curved canals (*De-Deus et al., 2013*). In our study all (100.0%) of the palatal canals were RFWL. All of these canals were straight canals or had only slight curvatures. Definitely these canals could be prepared to full WL most predictable. The buccal canals can be compared with the moderately curved canals, although also severely curved canals were included in our study. 97.2% of MB1 canals and 100.0% of DB canals in the 1°RCTx group were RFWL. Therefore, our clinical data reveal higher rates of RFWL than the *in-vitro* reference values. One clinical study compared the R25 and manual glide path preparation of MB2 root canals regarding RFWL in maxillary molars (*Zuolo, Carvalho & De-Deus, 2015*). Remarkably, in only 57.5% of the canals in the manual preparation group

full WL was reached, while in the R25 group this was possible in 85.6% of the canals (*Zuolo, Carvalho & De-Deus, 2015*). We found that 89.1% of the MB2 canals were RFWL with R25 in the 1°RCTx group. So our RFWL rate is about 3% higher. Most interestingly in the 2°RCTx group 96.1% of the MB2 canals were RFWL with R25 in our study.

To the best knowledge of the authors there is no clinical information available for root canal retreatments performed with Reciproc instruments with the attempt to waive glide path preparation. One clinical study describes the deformation and fracture rates of Reciproc instruments also in retreatment cases (*Plotino, Grande & Porciani, 2015*). However, it is not clearly stated if the retreatment cases were performed without glide path preparation. Moreover, this publication contains no information about the frequency of Reciproc to reach full WL. There are only *in-vitro* studies that evaluate the general possibility to use Reciproc instruments for endodontic retreatments (*De Souza et al., 2015*; *Marfisi et al., 2015*; *Zuolo et al., 2013*). All publications come to the conclusion, that Reciproc was the fastest system for retreatment. All studies found remaining filling material with all systems tested (*De Souza et al., 2015*; *Marfisi et al., 2015*; *Zuolo et al., 2013*). This is in concordance with our clinical experience, as it was necessary to control the cleanliness of the root canal walls under the DOM. All retreatment preparations were additionally fine finished with US instruments as described in the methods section to remove visible filling remnants.

The possibility to reach full WL with Reciproc in 2°RCTx cases was significantly lower in MB1 and DB canals than for 1°RCTx cases. The logistic regression model shows, that retreatment cases have a smaller chance for RFWL in MB1 canals, even if it is taken into account that we found more constricted cases in 2°RCTx. Additional difficulties may be pre-existent preparation faults as ledges, pre-existent via falsas that were impossible to correct and previously not properly approached MB1 canals that were instrumented in a wrong angle from disto-palatal instead of a straight line access. Moreover, it is also possible that Reciproc instruments caused a deviation from the original root canal trajectory or led to complete blockages of the canal. In some cases we did not find reasons for NRFWL, because it was also not possible to prepare a manual glide path. Apart from that the overall frequency of 89.3% of RFWL lies well between the 83.7% and 91.1% for 2°RCTx reported in the previously mentioned studies (*Ng, Mann & Gulabivala, 2011a*; *Ng, Mann & Gulabivala, 2011b*).

In our study in the palatal root canals, it was possible to reach full WL in every case regardless of 1°RCTx or 2°RCTx cases. Therefore, we conclude that these canals are the safest to be prepared without a glide path. We explain this by the fact that the palatal canals have less curvature, the biggest sizes, surface areas and volumes (*Peters et al., 2000*; *Wu et al., 2000*) compared to mesio- and distobuccal root canals.

In MB2/3 canals there was no statistical difference between the 1°RCTx and the 2°RCTx group. Interestingly the rate of RFWL was in the 1°RCTx group somewhat lower than in the 2°RCTx group. However, in the 1°RCTx group in some cases it was possible to obtain full WL with Reciproc after manual glide path preparation. For MB2/3 preparation, our logistic regression model identified constriction of the root canal as a negative predictor for RFWL. It is remarkable that this did not affect the other root canals. As there is no data available for the frequencies for RFWL in constricted root canals, we can only indirectly

guess that the substantially lower rate of RFWL with manual preparation in MB2/3 canals in the above mentioned study of *Zuolo, Carvalho & De-Deus (2015)* can be partially explained by the difficult preparation of obliterated root canals. Reciproc instruments overcome this problem in the way the manufacturer claims (*Yared, 2013a*). The positive predictor "convergent MB2/3 and MB1 canals" in preparation of MB2/3 canals for RFWL directly correlates with our clinical experience. Normally the preparation of MB2 canals is very predictable with a R25 if that canal is convergent to the MB1 canal.

In each of the treatment groups 1°RCTx and 2°RCTx one fracture of a R25 occurred. This is a very low fracture rate of overall 0.2% of the canals. Both fractures occurred in MB2 canals. In other studies higher fracture rates of up to 2.4% are reported with rotary NiTi instruments (*Wang et al., 2014*; *Wolcott et al., 2006*). The rate of 0.2% in our study lies well in the range of the clinically reported fracture rates of 0.21% (*Plotino, Grande & Porciani, 2015*) and 0.56% (*Zuolo, Carvalho & De-Deus, 2015*) for Reciproc.

In this study we observed a rate of 4.7% of apical perforations of the teeth treated. The overall rate of root perforations was reported in another study that assessed procedural errors of endodontic treatments with 4.5% (*Silva et al., 2012*). The rate of root perforations in posterior maxillary teeth was reported even higher with 5.8%. Therefore, the rates of our study are comparable with these studies. A limitation of our study is the correct diagnosis of root perforations. All perforations were apical perforations and were assessed with two-dimensional X-rays. Therefore, the information on the actual three-dimensional shape of the original trajectory of the root canal is incomplete and accordingly also the assessment of the real rate of perforations. In this context, the original anatomy of the root canal of course will influence the event of perforations. Especially in cases with abrupt apical curvatures or complete apical constrictions of the canal, even very flexible root canal instruments sometimes cannot follow the original trajectory and may cause undetected perforations. Because this study was a retrospective study, it was not possible to further evaluate the influence of the original root canal anatomy on RFWL with Reciproc instruments.

It was possible to prepare the vast majority of endodontic primary and retreatment cases with Reciproc to full working length without prior glide path preparation. Reciproc instruments contribute in a highly universal way to the armamentarium of the endodontic clinician. Within the limitations of this study, we suggest waiving the traditional glide path preparation with Reciproc instruments.

### Funding

The authors received no funding for this work.

### Competing Interests

The authors declare there are no competing interests.

## Author Contributions

- Andreas Bartols conceived and designed the experiments, performed the experiments, analyzed the data, contributed reagents/materials/analysis tools, wrote the paper, prepared figures and/or tables, reviewed drafts of the paper.
- Bernt-Peter Robra analyzed the data, wrote the paper, prepared figures and/or tables, reviewed drafts of the paper.
- Winfried Walther contributed reagents/materials/analysis tools, wrote the paper, reviewed drafts of the paper.

## Human Ethics

The following information was supplied relating to ethical approvals (i.e., approving body and any reference numbers):

The Institutional Review Board of the Baden-Württemberg Medical Council approved this study (AZ: F-2016-031-z).

## Data Availability

The raw data has been supplied as a Supplementary File.

## Supplemental Information

Supplemental information for this article can be found online at http://dx.doi.org/10.7717/peerj.3583#supplemental-information.

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
