# Peer review of "The ability of Reciproc instruments to reach full working length without glide path preparation: a clinical retrospective study"

_PeerJ, doi:10.7717/peerj.3583_

## Round 0.1 · original submission · Major Revisions

Your manuscript has been revised by two subject experts who have made several comments and suggestions. An annotated copy of the manuscript is also attached with some formatting-grammatical suggestions. We look forward to receiving your resubmission.

Reviewer 1 ·

Basic reporting

The English language needs to be revised and improved, mainly for the introduction and discussion sections.

Some concepts require revision:

-Definition of MB2 as an “accessory canal”
Lines 32, 46, 60, 116, 172, 212, 216 – The high incidence of MB2 is well known and described in the literature. According to Vertucci (Endodontic Topics 2005, 10, 3–29)
“An accessory canal is any branch of the main pulp canal or chamber that communicates with the external surface of the root”. Therefore, the MB2 or any other extra canal that presents its own orifice should not be classified as accessory.

-Use of Reciproc
The aim of the paper is to report the clinical use of Reciproc without glide path. So, it is important to explain for the readers the concepts of glide path and also provide a quick background of Reciproc instrument, which are meant to be single-file technique in a reciprocant motion. Also, about this matter, the methods states that “Reciproc instruments were used strictly according to the manufacturers recommendations (Lines 120-123)”. However, in the Line 137 it is written that canals were further enlarged a size #35. Please better explain the rationale of this last procedure and also provide the references.

Experimental design

Overall the experimental design is clear and adequate.
However, some points are listed bellow:

Line 92. Is the operator a GP or endo specialist? This information is relevant to acknowledge the expertise of the operator.

Line 100. What is the definition of obliterated? Is this the pulp chamber, narrow canals, lack of apical patency? Also, according to Table 3 it seems that obliterated was referenced per canal not per tooth. I think is better to clarify the definition and if it was determined per tooth or per canal.
Additionally, I noticed that for mb2 it is clear that obliteration is more relevant for the RFWL than the type of treatment. However, it does not specify if the canals in the retreatment were actually missed canals, therefore they were “prepared for the first time”. Also, how many of those were the 5th canal? I assume that cases that the combination of a 5th canal in retreatment could also indicate a missed canal, therefore helps to explain why the RFWL was more frequent in retreatment than in 1o RCTx.
Again, for Mb1, it seems that the number of NRFWL is correlated to the type of treatment, but how many retreatments presented an “obliterated” canal?
Thus, I think is better to clarify the criteria for classifying the canals as obliterated. Also, maybe an if the analysis could consider the combination of 2o RCTx AND obliterated, then it would be more clear if the problem is the filling removal or the obliteration itself.

Line 101-102. Table 1 lists 9 cases with a 5th canal. How was this canal classified? I think they were included as mb2, however, a 5th canal should not be considered as mb2, (because usually one of the other 4 is already the mb2)

Line 129. In the methods is stated that patency was reached and used the apex locator with a #10 file in all cases. If the Reciproc didn’t achieve the WL, you performed a glide path. But why did the glide path started with a #6 or #8 if your WL was already reached with a #10? Please explain better if those steps were made to obtain patency or glide path.

Line 132. There was any case in which patency was not achieved?

Line 133. It would be interesting to list all cases that a larger file was used, as well as if in any case it was used only the R20, because this indicates that the anatomy is a predictor of reaching or not the WL. For example, you might have a very large MB1 in a young patient in which is easy to reach the WL because the canal presents a bigger apical diameter.

Validity of the findings

About the Results:
Line 181-185. In five cases the Reciproc didn’t reach the WL and in all cases it was not possible to re-gain the original trajectory ( the report says 2 cases were incompletely prepared and 3 were via falsa). It seems that the 3 cases of via falsa should be reported as apical transportation or apical perforation. The other 2 cases seem like apical ledge or blockage or patency lost due to the use of the Reciproc, unless those cases were previously blocked or non-patent. If so, this should also be reported and explained.

Lines 189-190. It seems that the sentences “in five out of eight…” and “also the manual…” should be merged for a better understanding.
Line 191. Those 3 cases should be reported as apical transportation.
Line 191-192. It is not clear if it was treated surgically only the 3 cases (via falsa) or all the 8 cases in which Reciproc was NRFWL

In the Discussion section:

Lines 276-279. In the “methods section” it is described that the WL and patency were obtained in all cases with a #10 file. If in 98 canals the #10 was not reaching the WL, how the WL was even detected? Please explain those case in the methods and also explain better in the discussion. Additionally, the assumption that the fact a #10 does not reach the WL would automatically lead to a what is stated in line 279 (would have been prepared incompletely) is misleading. Since you didn’t use any other instrument to establish a comparison, you can state that using R25 facilitate a #10 of reaching the WL in 67.3% of the cases, but in the other cases additional steps/files/glide path had to be used (assuming that in those cases you performed what was described in methods section, about using files 6,8,10 and 15)

Lines 325-326. Once again, if it was not possible to reach WL with Reciproc, and also the manual files are not reaching the WL, would this be an indicator that the R25 instrument caused a deviation of the trajectory or because this was a ledge caused by the prior root canal treatment?

Line 328-330. This whole paragraph seems to be addressing only the retreatment cases, but the last 2 sentences about palatal canals seem to address also 1oRCTx. Please clarify this. Also, the explanation of palatals canals being "nearly straight" should be referenced or better discussed because the same percentage occurred for DB canals in 1oRCTx. So, the tendency of "being straight" could be the main, but not the only factor. Is it important to address also facts such as diameter, cross-section, furcations, dentinal triangle, etc.

Additional comments

This article seems to bring relevant information about clinical use of Reciproc.
However, the article should provide a better background for the readers that are not familiar with the concept of Reciproc. This instrument is very peculiar and is still the only "reciprocant" instrument that is advised to be used without a prior glide path. However, even if the clinical details of the sequence of instruments were published previously, it is important to describe them also in this paper, mainly for the cases that were classified as NRFWL.

Reviewer 2 ·

Basic reporting

English should be checked by an editting service.
There is no hypotheses.

Experimental design

No comment

Validity of the findings

No comment

Additional comments

Thank you for your submission to PeerJ journal. The manuscript is overall well-written. There are some minor suggetion on the pdf document. You can see the comments on the pdf file.

Annotated reviews are not available for download in order to protect the identity of reviewers who chose to remain anonymous.

---

## Round 0.2 · Minor Revisions

Please address the concern raised by Reviewer 1. We will not send it out for a review again as long as you are able to specifically address this issue.

Reviewer 1 ·

Basic reporting

The English language still needs revision.

Experimental design

Thank you for the explanation of what "obliterated canal" means. However, I think that this term is not appropriated.
As you explained, you defined obliteration when the 6 file could not easily advance in the apical portion of the canal. It seems that this would be better denominated as "constricted or narrow canals".
According to some authors, pulp space obliteration refers to the process of calcification that occurs from the crown to the apex ( See McCabe & Dummer 2012, Int Endod J. , Pulp canal obliteration: an endodontic diagnosis and treatment challenge.) Therefore, it does not comply with the description you provided, unless you can quote a reference that explains your concept.

Validity of the findings

I would like to express my concerns regarding your comments about the study findings.
I totally agree that research should answer clinically relevant questions. However, the fact that you can RFWL with Reciproc does not mean that you are in the correct original trajectory. You reported only few cases in which you detected perforation or via falsa and this is probably related to the fact you used only 2D evaluation. So, I do not agree that your methods overestimated NRFWL, but you have a bias of method. As I suggested, if you could report the cases in which the canal was already originally larger (those canals the glide path would not be needed anyway), the results would be more valuable.
If you are not able to provide this data, at least would be important to include those limitations of methods in the discussion, also mention the fact that the original anatomy (that was not checked) could also play a role.

Also, the conclusion should include "within the limitations of the study".

Reviewer 2 ·

Basic reporting

No comment

Experimental design

No comment

Validity of the findings

No comment

Additional comments

No comment

---

## Round 0.3 · accepted · Accept

The reviewer comments are appropriately addressed.